# Transcriptional Analysis of a Tripartite Interaction Between Maize (*Zea mays*, L.) Roots Inoculated with the Pathogenic Fungus *Fusarium verticillioides* and Its Bacterial Control Agent *Bacillus cereus sensu lato* Strain *B*25

**DOI:** 10.3390/plants14233661

**Published:** 2025-12-01

**Authors:** Paúl Alán Báez-Astorga, Abraham Cruz-Mendívil, Juan Luis Figueroa-Castro, Itzel Guadalupe López-Soto, Jesús Eduardo Cazares-Álvarez, Josefat Gregorio-Jorge, Carlos Ligne Calderón-Vázquez, Ignacio Eduardo Maldonado-Mendoza

**Affiliations:** 1SECIHTI—Instituto Politécnico Nacional, CIIDIR Unidad Sinaloa, Guasave 81100, SIN, Mexico; a_lan6@hotmail.com (P.A.B.-A.); abcruz@outlook.com (A.C.-M.); eduardo.cazaresalvarez@gmail.com (J.E.C.-Á.); 2Universidad Autónoma de Occidente (UAdeO), Unidad Regional Guasave, Departamento de Ciencias Naturales y Exactas, Guasave 81044, SIN, Mexico; juanfc2197@gmail.com (J.L.F.-C.); itzela0273@gmail.com (I.G.L.-S.); 3SECIHTI—Comisión Nacional del Agua, Ciudad de México 03940, Mexico; josefatgregorioj@gmail.com; 4Departamento de Biotecnología Agrícola, Centro Interdisciplinario de Investigación para el Desarrollo Integral Regional (CIIDIR) Unidad Sinaloa, Instituto Politécnico Nacional, Guasave 81100, SIN, Mexico; cligne@gmail.com

**Keywords:** RNA-seq, maize transcriptome, endophyte, *Bacillus cereus*, *Fusarium verticillioides*, differentially expressed genes, biological control, plant–microbe interactions

## Abstract

One open question regarding plant–microbe interactions is how a plant interacts molecularly with both a beneficial microbe and a pathogenic fungus. This study used RNA-seq to investigate molecular responses in maize roots during a tripartite interaction with the fungal pathogen *Fusarium verticillioides* (*Fv*), which causes stalk, ear, and root rot, and the endophytic biocontrol agent *Bacillus cereus sensu lato B*25, known to suppress *Fv* and promote plant growth. Roots of seven-day-old maize inoculated with *Fv* (*Zm*-*Fv*), *B*25 (*Zm*-*B*25), and co-inoculated (*Zm*-*Fv*-*B*25) were compared to uninoculated control (*Zm*). Differential Gene Expression (DEG), Gene Ontology (GO) and KEGG analysis revealed distinct molecular responses. *Fv* suppressed plant pathways related to DNA and protein synthesis and impaired root development. In contrast, *B*25 triggered defense priming and growth-related responses. In the co-inoculation experiment (*Zm*-*B*25-*Fv*), upregulated DEGs were associated with both defense-related metabolic pathways, including jasmonic acid signaling and secondary metabolite biosynthesis, and genes involved in plant growth processes. Co-expression networks using *Arabidopsis* orthologs supported the induction of defense- and growth and development-related genes. This study is the first RNA-seq analysis of maize root molecular responses during the tripartite interaction with a fungal pathogen and its bacterial biocontrol agent, providing new directions for further research to understand the detailed molecular mechanisms underlying this interaction fully.

## 1. Introduction

Currently, over a billion tons of maize are produced every year, making it the most abundant cereal crop globally [1]. Maize is susceptible to different fungal pathogens, with *Fusarium verticillioides* (*Fv*) being the most common cause of stalk, ear, and root rot (SERR) [2,3]. *Fv* is a versatile fungus that can infect maize either through seeds (vertical transmission) or by environmental causes such as dissemination through the soil, air, or natural wounds inflicted by herbivores (horizontal transmission). As a hemibiotroph, *Fv* can either live harmlessly inside maize as a symptomless endophyte or cause disease as a necrotrophic pathogen [4,5]. *Fv* not only affects maize yield but also grain quality, due to mycotoxin contamination [6]. 

Due to increasing global concerns about environmental impacts and the human health effects of synthetic agricultural chemicals (such as pesticides, fungicides, insecticides, and herbicides), there is a strong drive to identify sustainable ways to manage crop diseases [7]. Among such alternatives, the use of biological control agents (BCAs) is not only efficient in controlling target pests and diseases, but they also improve plant growth, development, and ultimately crop yields [8,9]. Bacterial endophytes from the genera *Bacillus*, *Paenibacillus*, *Burkholderia*, *Pseudomonas*, and *Staphylococcus* have been successfully employed as BCAs to effectively control SERR caused by *Fv* on maize [10,11,12,13]. Endophytes are non-pathogenic microorganisms that inhabit the internal parts of plants for at least part of their life cycle and are generally considered beneficial due to their plant growth-promoting and protective properties [14,15]. These beneficial microorganisms use several mechanisms to promote plant growth and act as biocontrol agents. To enhance plant growth, endophytes can fix nitrogen, solubilize phosphate and potassium, produce siderophores, and synthesize various phytohormones such as auxin, cytokinin, and gibberellin [14]. On the other hand, these endophytes can employ direct and indirect mechanisms of antagonism for biocontrol. Indirect antagonism in biocontrol works by priming the host plant defenses, making them react faster and more powerfully when pathogens attack. In contrast, direct antagonism is achieved through mechanisms like hyperparasitism, predation, and the production of lytic enzymes and antibiotics [14,16].

The advantage endophytes have over other biocontrol agents is their ability to colonize the plant internal tissues, which allows them to share the same ecological niche as some phytopathogens within the plant host, increasing their value as biocontrol candidates [14,15]. As an example of this, the maize endophytic biocontrol agent *Bacillus cereus* strain *B*25 colonizes the maize root vasculature [17], just like the phytopathogen *Fv* [4].

*B*25 has demonstrated effective control of *Fv* through in vitro, greenhouse, and field trials [9,13,18]. The genome of *B*25 harbors genes for several antagonistic mechanisms against fungal pathogens, including the production of lytic enzymes (chitinases A and B, chitosanase, and glycoside hydrolase), siderophores (petrobactin and bacillibactin), antibiotics (surfactin), and biofilm production [19]. These mechanisms have been biochemically and molecularly characterized and appear to be involved in the direct biocontrol of *B*25 over *Fv*, as all of them are induced during their direct interaction in vitro [13,20,21,22]. In addition, *B*25 reduces fumonisin levels and increases maize yield [9]. 

Although we understand the antagonism mechanisms used by *B*25 to control *Fv* (bacteria–fungus interaction) and the maize physiological responses when inoculated with *B*25 and challenged with *Fv*, the molecular mechanisms involved in the maize-*B*25-*Fv* tripartite interaction have not been investigated at the transcriptional level. To understand this tripartite interaction and capture the molecular communication (mRNA) of microorganisms in their proportions relative to maize, this work focuses on the molecular responses of maize in its interactions with the bacterial biocontrol agent and the fungal pathogen.

Here, we address the question of how a plant could simultaneously interact with a phytopathogenic fungus and a beneficial organism. Because studying interactions among three organisms is particularly challenging, we developed a model of the tripartite interaction between maize, the phytopathogenic fungus *Fv* (which causes SERR), and its biological control agent, the endophytic bacterium *B*25. RNA-seq is a method for understanding the molecular mechanisms underlying specific host–microorganism interactions [23]. By analyzing the maize transcriptome in response to inoculation with (1) the biocontrol endophyte *B*25 (*Zm-B*25), (2) the fungal pathogen *Fv* (*Zm-Fv*), and (3) both *B*25 and *Fv* (*Zm-B*25*-Fv*), we have identified key genes and molecular mechanisms involved in the maize–*B*25–*Fv* (*Zm-B*25*-Fv*) tripartite interaction.

## 2. Results

### 2.1. B25 Mediated the Biocontrol of Fv and Enhanced Maize Plant Growth in the Tripartite Interaction

To confirm that the rolled paper seedling bioassay enabled initial host plant colonization and to assess the plant’s physiological responses to each organism in bi- and tripartite interactions, we microbiologically verified colonization and measured plantlet biomass and length. Microbial colonization was visually confirmed on disinfected maize roots (Appendix A), with no growth on controls (*Zm*). Bipartite associations (*Zm-B*25 and *Zm-Fv*) showed greater microbial growth than the tripartite interaction (*Zm-B*25*-Fv*). *Fv* inoculation (*Zm*-*Fv*) significantly reduced root length and fresh weight compared to other treatments (*Zm*, *Zm*-*B*25, *Zm*-*B*25-*Fv*) (Table 1, Appendix A). However, co-inoculation with *Fv* and *B*25 (*Zm*-*B*25-*Fv*) resulted in a significant increase in leaf length and fresh weight compared with all other conditions (*Zm*, *Zm*-*B*25, *Zm*-*Fv*). 

### 2.2. RNA Sequencing of the Tripartite Assay

More than 162 million raw paired-end reads were obtained from the sequencing of the whole tripartite experiment, with an average of 20 million reads per sample (Table 2). After quality trimming (>Q20), an average of 98.49% of the reads were retained per library. Additionally, 95.93% of these filtered reads also achieved a quality score of >Q30. The mean values for unique and multiple-mapped reads were 17,703,196 (87.28%) and 758,808 (3.71%), respectively.

Principal component analysis (PCA) was performed to confirm consistency across biological replicates. One biological replicate from each of the four conditions (*Zm*, *Zm-B*25, *Zm-Fv*, and *Zm-B*25*-Fv*) showed atypical behavior and was thus removed, leaving two biological replicates per condition, as described in the Materials and Methods Section 4.5 (Appendix A).

### 2.3. Global Transcriptome Profiles of Maize Roots from the Tripartite Assay

Expressed genes (EGs) were defined as those averaging at least 10 read counts across two biological replicates. Maize roots in the tripartite assay expressed 24,628 out of 44,303 predicted genes from the *Zea mays* B73 v5 genome, representing 55.59% of the genome (Table 3). Inoculation of maize roots with *B*25 and/or *Fv* induced the expression of a higher percentage of the genome compared to the control (*Zm*). Interestingly, maize root colonization by *B*25 (*Zm-B*25) resulted in the highest percentage of gene expression, accounting for 57.36% (25,412 genes) of the genome. A comparative analysis across all conditions revealed a core of 22,576 (86.6%) common EGs. However, individual conditions showed distinct profiles: *Zm-B*25 exhibited the largest number of unique EGs (454, 1.7%), followed by *Zm* (254, 1%), *Zm*-*Fv* (215, 0.8%), and *Zm-B*25-*Fv* (189, 0.7%) (Figure 1).

Unique EGs from the interaction conditions were subjected to GO analysis. The results showed 11, 8, and 4 overrepresented terms for *Zm-B*25, *Zm-Fv*, and *Zm-B*25*-Fv*, respectively (Appendix A). Overrepresented terms within the unique EGs of *Zm-B*25 were associated with transcription regulation and negative regulation of catalytic activity. Additionally, the response to auxin was also overrepresented in this interaction. On the other hand, overrepresented terms within the set of unique EGs in *Zm-Fv* were related to photosynthesis regulation, primary metabolic processes, cell fate determination, and transcriptional regulation. Finally, the terms overrepresented in *Zm-B*25*-Fv* unique EGs were associated with actin cytoskeleton restructuring and growth (Appendix A).

### 2.4. Differentially Expressed Genes Within and Among Maize Interactions

The number of DEGs for the *Zm-B*25, *Zm-Fv*, and *Zm-B*25*-Fv* conditions was 4331, 5473, and 2334, respectively (Table 4, Appendix A). The three interactions shared a common core of 1948 DEGs (Figure 2). The tripartite *Zm-B*25*-Fv* interaction showed the fewest unique DEGs (78), in contrast to the bipartite *Zm-B*25 (653) and *Zm-Fv* (1513) interactions, which exhibited more unique DEGs. Furthermore, *Zm-B*25*-Fv* shared 295 DEGs exclusively with the *Zm-Fv* interaction and only 13 with the *Zm-B*25 interaction, while *Zm-B*25 and *Zm-Fv* shared a substantial 1717 DEGs. 

In general, the proportion of upregulated genes (>74%) was higher than that of downregulated genes across all conditions (Table 4). Bacterial (*Zm-B*25) and bacterial-pathogen (*Zm-B*25*-Fv*) inoculation of maize resulted in gene expression patterns with approximately 89% of genes upregulated and only 10–11% downregulated. In contrast, 25.64% of genes were repressed in *Zm-Fv* (Table 4).

By analyzing the number of up- and downregulated genes (both unique and shared) within each interaction, we gained a comprehensive understanding of the transcriptome’s overall behavior in this study. Whereas 6 out of 7 Venn diagram sections displayed a common trend of more upregulated than downregulated genes, unique DEGs in *Zm*-*Fv* showed the opposite trend. In *Zm*-*Fv*, downregulated genes were more numerous (972; 64.24%) than upregulated genes (541; 35.75%). In summary, *Zm*-*Fv* had the highest number of DEGs, accumulating most of the unique DEGs as well as those shared with *Zm*-*B*25 and *Zm*-*B*25-*Fv*.

### 2.5. Overrepresentation Analysis: A General Approach

GO analysis revealed significant overrepresentation (adjusted *p*-value ≤ 0.05) of 64 categories in the *Zm-B*25*-Fv* interaction, 90 in the *Zm-B*25 interaction, and 60 in the *Zm-Fv* interaction (Figure 3; Appendix A). The *Zm-Fv* interaction had the highest number of DEGs (5473), but the fewest overrepresented categories. Overall, the overrepresented categories belonged mainly to Molecular Function (MF) with 54, followed by Biological Process (BP) with 38, and Cellular Compartment (CC) with 10. The three conditions shared 42 overrepresented categories. Notably, *Zm-B*25 exhibited the highest number of unique overrepresented categories with 27, followed by *Zm-B*25*-Fv* with 8, and *Zm-Fv* with 5 (Figure 3).

Among the eight unique and overrepresented GO terms for *Zm-B*25*-Fv*, four corresponded to the BP category defense response (GO:0006952), and response to jasmonic acid (GO:0009753) appeared together with suberin biosynthetic process (GO:0010345) and transmembrane transport (GO:0055085) (Appendix A). The myosin II complex (GO:0016460) was the sole overrepresented term identified in the CC category. In contrast, nutrient reservoir activity (GO:0045735), protein phosphatase inhibitor activity (GO:0004864) and manganese ion binding (GO:0030145) were found in the MF category (Appendix A).

For the *Zm*-*B*25 interaction, the unique overrepresented GO terms in the BP category included those related to plant growth and development, such as aromatic compound biosynthetic process (GO:0019438), phenylpropanoid and carbohydrate metabolic process (GO:0009698 and GO:0005975), and plant-type secondary cell wall biogenesis (GO:0009834). In the MF category, a diverse set of overrepresented terms were identified, primarily related to transcription regulation (GO:0000981), catalysis of O-glycosil compounds (GO:001616 and GO:0042973), and transferase activity (ubiquitin (GO:0061630), as well as the pentosyl (GO:0016763), glycosil (GO:0016757), and acyl groups (GO:1990538), oxido-reductase activity (GO:0016616, GO:0016618, GO:0016701, and GO:0051213), calcium-dependent phospholipase A2 activity (GO:0047498), structural constituent of chromatin (GO:0030527), and ammonia-lyase activity (GO:0016841). The overrepresented terms for the CC category included extracellular space (GO:0005615), nucleosome (GO:0000786), chromatin (GO:0000785), and Golgi transport complex (GO:0017119). 

In the *Zm-Fv* interaction, four out of the five unique overrepresented GO terms were associated with DNA replication (GO:0003677, GO:0017116, GO:0006270, and GO:0045910).

Interestingly, *Zm-B*25 and *Zm-Fv* shared 10 overrepresented GO terms (Figure 3, brown box). Several terms stood out among these overrepresented GO terms, including those associated with transcription (GO:0003700, GO:0006355, GO:0043565), oxylipin biosynthesis (GO:0031408), and the systemic acquired resistance response (GO:0009627).

### 2.6. KEGG Analysis 

KEGG analysis revealed 13, 19, and 20 overrepresented pathways in the *Zm-B*25*-Fv*, *Zm-B*25, and *Zm-Fv* interactions, respectively (Figure 4, Appendix A). Only *Zm-B*25 (5) and *Zm-Fv* (6) exhibited uniquely overrepresented KEGG pathways. The five unique pathways identified in *Zm-B*25 were associated with plant growth and development (zma00905: brassinosteroid biosynthesis), signaling (zma00590: arachidonic acid metabolism, zma04016: MAPK signaling pathway—plant), amino acid metabolism (zma00250: alanine, aspartate, and glutamate metabolism), and production of secondary metabolites (zma00944: flavone and flavonol biosynthesis). On the other hand, five out of the six unique overrepresented KEGG pathways in the *Zm-Fv* interaction were related to DNA replication and maintenance, including DNA replication (zma03030), homologous recombination (zma03440), base excision repair (zma03410), and non-homologous end-joining (zma03450). Additionally, ribosome biogenesis in eukaryotes (zma03008) was a unique overrepresented pathway. The remaining unique KEGG pathway was motor proteins (zma04814) (Figure 4, Appendix A).

The 13 KEGG pathways overrepresented across all conditions were primarily linked to the biosynthesis of various secondary metabolites and to the metabolism of glutathione, linoleic acid, and galactose. The starch and sucrose metabolism (zma00500) pathway was only altered when maize interacted with either *B*25 or *Fv* individually, but not when both microorganisms were present within the plant. 

### 2.7. Functional Gene Association Networks: Enriched Biological Processes Using A. thaliana Orthologous Genes

Although the GO overrepresentation analysis based on maize gene annotations from BioMart provided insight into each interaction as described above, an additional GO enrichment analysis was conducted using *A. thaliana* orthologous genes (Appendix A, Appendix A) to examine the governing mechanisms behind each interaction. Using up- and downregulated genes from each interaction and their intersection (Figure 2), we built gene network associations to understand the functional relationships among DEGs. Appendix A shows the main results obtained from the analysis of NRGIs using String.

Analysis of NRGIs revealed diverse networks for each maize interaction (Appendix A). Even though the *Zm-B*25*-Fv*/*Zm-Fv*/*Zm-B*25 (UP) gene set harbored the highest number of nodes (874), the *Zm-Fv* (DOWN) gene set (678) exhibited a higher number of edges (5866) than the former one (2560). Gene sets that produced a main network with fewer than 200 nodes were either expanded via String or submitted to Genemania as a complementary approach to identify enriched pathways. Significant associations for *Zm-B*25*-Fv* (UP), *Zm-B*25*-Fv/Zm-B*25 (UP), and *Zm-B*25*-Fv/Zm-B*25 (DOWN) were only detected after the String network expansion and Genemania analysis (Appendix A).

Overall, rather than evaluating individual genes, obtaining BPs from gene association networks enabled us to clearly and comprehensively identify the most prominent genes associated with each maize interaction (Figure 5). For all cases in which *Zm-Fv* appeared (4 out of 7; see sections *Zm-B*25*-Fv*/*Zm-Fv*/*Zm-B*25, *Zm-Fv*/*Zm-B*25, *Zm-Fv* and *Zm-B*25*-Fv*/*Zm-Fv*), the response to the stress-related stimulus (chemical) was the prevalent process among upregulated genes. In contrast, the process of DNA-related metabolism stood out among downregulated genes (Appendix A). Both processes covered 5473 DEGs (88.03%) out of the total 6217 DEGs. Even though the *Zm-B*25*-Fv*/*Zm-Fv*/*Zm-B*25, *Zm-Fv*/*Zm-B*25, *Zm-Fv*, and *Zm-B*25*-Fv*/*Zm-Fv* interactions shared similar BPs within upregulated genes, *Zm-B*25*-Fv*/*Zm-Fv* exhibited exclusive BPs such as photosynthesis, generation of precursor metabolites and energy, and small molecule metabolic process, reinforcing the previous observation that *Zm-B*25*-Fv*/*Zm-Fv* differed slightly from the other interaction sections in which *Zm-Fv* was involved.

The remaining 744 DEGs (11.97%) clustered into the last three sections (*Zm-B*25, *Zm-B*25*-Fv*, and *Zm-B*25*-Fv*/*Zm-B*25). BPs associated with upregulated genes were the most contrasting compared to 88.03% of DEGs. For example, *Zm-B*25 harbored exclusive BPs such as carbohydrate metabolic process, cell wall organization or biogenesis, and regulation of macromolecule biosynthetic process, whereas *Zm-B*25*-Fv*/*Zm-B*25 showed overrepresented processes such as phospholipid biosynthetic process and inositol metabolic process (Appendix A). On the other hand, *Zm-B*25*-Fv* featured BPs such as ribosome biogenesis, gene expression, rRNA processing, cellular nitrogen compound metabolic process, ribosome assembly, macromolecule metabolic process, and translation. Finally, it is important to note that *Zm-B*25 and *Zm-B*25*-Fv* also showed DNA-related metabolic processes among their downregulated genes, which differed sharply from *Zm-B*25*-Fv*/*Zm-B*25, which featured BPs such as phosphatidylinositol-3-phosphate biosynthetic process, protein targeting mitochondrion, protein transmembrane transport, vacuole organization, photomorphogenesis, and autophagy-related processes (Appendix A). Functional gene association networks, therefore, provide an overall insight into the transcriptomic behavior covering each section and intersection of maize interactions.

### 2.8. Gene Interaction Networks Based on Co-Expression

Gene associations often yield highly connected networks with little informative power. Therefore, clustering them through algorithms allows for a better interpretation (Appendix A). The most significant identified clusters are highly relevant to the main findings of the original network. Therefore, to gain more insight into previously described functional networks (Appendix A), Markov clustering (MCL) was performed based on co-expression using *A. thaliana* orthologs. Interactions involving *Zm-B*25*-Fv*, *Zm-B*25, and *Zm-Fv*, as well as their corresponding intersections (*Zm-B*25*-Fv/Zm-B*25, *Zm-B*25-*Fv*/*Zm-Fv*, and *Zm-B*25/*Zm-Fv*) were considered for this analysis.

The *Zm-B*25/*Zm-Fv* intersection harbored a central cluster of upregulated genes related to stress response (response to chemicals and chitin), cell wall remodeling (biosynthesis of secondary cell wall), phenylpropanoid metabolic process, oxoacid metabolic process, hormone-mediated signaling pathway (auxin), and photosynthesis (Appendix A). In *Zm-B*25, chromatin remodeling, gene expression, protein metabolic process (translation), nucleocytoplasmic trafficking of RNAs and proteins, phenylpropanoid-related metabolic process, hormone-mediated signaling pathway (auxin), protein phosphorylation (involved in growth and development), and thermotolerance were all exclusive to this interaction. BPs exclusive to the *Zm-Fv* interaction were found, such as response to external stimuli (bacteria and fungi), glycolysis, photosynthesis, carboxylic acid metabolic process (oxidative phosphorylation), very-long-chain fatty acid (VLCFA) biosynthesis, transition metal ion transport, pigment metabolic process, nitrate assimilation, cellular detoxification, proteasomal protein catabolic process, and circadian rhythm (Appendix A). Noteworthy, the cell wall biogenesis process, specifically the biosynthesis of secondary cell wall, was shared between *Zm-B*25 and *Zm-Fv*.

*Zm-B*25*-Fv* exclusive BPs, as well as BPs shared with *Zm-B*25 and *Zm-Fv* were analyzed. First, upregulated *Zm-B*25*-Fv* genes exhibited overlapping BPs with *Zm-B*25, such as lipid metabolic processes (phospholipid biosynthetic process and inositol metabolic process), L-ascorbic acid biosynthetic process, transcription regulation of SA- (positive) and JA- (negative) related genes, and melatonin biosynthetic process. Secondly, the BPs shared with *Zm-Fv* included cellular response to oxidative stress (chemical), generation of precursor metabolites and energy (photosynthesis and glycolysis), cellular aldehyde metabolic process (related to oxidative stress tolerance), cellular detoxification, mRNA polyadenylation, VLCFA biosynthesis, and plant hormone signal transduction (ABA). Altogether, the above-described BPs allowed us to identify the exclusive BPs corresponding to *Zm-B*25*-Fv*, including ribosome biogenesis (ribosome assembly), actin filament depolymerization, and polysaccharide catabolic process (stress-induced starch degradation) (Appendix A). On the other hand, protein metabolic (translation) and lipid metabolic (phospholipid biosynthetic process) processes found in the *Zm-B*25*-Fv* interaction were also present in *Zm-B*25 and *Zm-B*25*-Fv/Zm-B*25, respectively. However, it is important to note that the NRGIs involved in both BPs were different from those between *Zm-B*25 and *Zm-B*25*-Fv*, or between *Zm-B*25*-Fv* and *Zm-B*25/*Zm-B*25*-Fv*. 

Although clusters formed by downregulated genes involved DNA metabolic processes in general, peculiarities in their interactions and intersections could be observed when compared with those of cluster-associated BPs of upregulated genes. For example, whereas *Zm-Fv* exhibited BPs associated with response to heat and protein folding in their downregulated genes, such BPs were among the upregulated *Zm-B*25 genes. Furthermore, BPs associated with the process of translation (ribosome biogenesis, rRNA processing, and ncRNA metabolic process) identified among the downregulated genes of *Zm-Fv* and *Zm-B*25 (as well as the *Zm-B*25/*Zm-Fv* intersection) were found among the upregulated genes of *Zm-B*25*-Fv*.

In addition to the identified exclusive BPs, other processes were also associated with the *Zm-B*25*-Fv* tripartite interaction, such as cellular response to oxidative stress and its corresponding counteracting mechanisms (transcription regulation of hormone-related genes [SA and JA], plant hormone signal transduction [ABA], cellular aldehyde metabolic process, and cellular detoxification involving the glutathione-ascorbic acid redox cycle). Changes occurring in the plasma membrane and plant cuticle are suggested by the identification of lipid metabolic processes (phospholipid biosynthetic process, inositol metabolic process, and VLCFAs). Finally, the generation of precursor metabolites and energy (photosynthesis, glycolysis, and stress-induced starch degradation) also emerged as important processes occurring in this interaction. In summary, the *Zm-B*25*-Fv* interaction seems to harbor BPs related to plant growth and development (gene expression, RNA metabolic process, ribosome biogenesis, translation, actin filament depolymerization, photosynthesis and glycolysis), as well as BPs associated with plant protection triggered by pathogens (changes in plasma membrane and cuticle, as well as cellular response to oxidative stress), which may help explain the *B*25-mediated maize protection against *Fv*.

### 2.9. Validation of RNA Sequencing by qRT-PCR

To confirm our RNA-seq data analysis, RT-qPCR analysis was conducted to validate the expression patterns of selected genes using a different RNA expression measurement technique. Eleven unique DEGs were selected based on their FC (high and/or low) and the number of reads (≥30 reads per biological replicate) from the interactions, and their expression profiles were evaluated by qRT-PCR. As indicated in Table 5, the FC values obtained by qRT-PCR analysis showed expression trends comparable to those in the RNA-seq data, thus confirming the RNA-seq data’s feasibility.

## 3. Discussion

Bacterial endophytes, which are abundant in plants, are currently being extensively studied for their plant growth-promoting effects and their ability to protect plants against diseases. A deeper understanding of not only the bacterial endophyte-host plant interaction, but also the fungal pathogen-host plant interaction and the tripartite interaction between the bacterial endophyte, the fungal pathogen, and the host plant can enhance the use of bacterial endophytes to overcome plant disease and improve agricultural production [15,16,24].

Previous studies have demonstrated the potential of *B*25 to protect maize plants against *Fv* and to promote their growth at early [13,18] and late growth stages [9]. To better understand the molecular basis of the biocontrol and plant growth promotion induced by *B*25 on maize against the fungal pathogen *Fv*, we analyzed the transcriptomic changes in maize roots obtained at seven days after seed inoculation with *B*25 and *Fv*, either individually or combined. The expression patterns of several DEGs were validated in this work (Table 5), complementing a previous report that validated 10 induced maize chitinase DEGs from the *Zm-Fv* condition [25]. To the best of our knowledge, this study represents the first report of a maize root transcriptome in response to dual inoculation with two endophytes: a bacterial biocontrol agent and a fungal pathogen. 

### 3.1. Plant–Bacteria Interaction (Zm-B25): B25 Is Recognized by Maize as a Beneficial Microorganism

A core of 1961 DEGs was shared among all conditions. Of these, only the gene *Zm00001eb141540* (geb1-glucan endo-1,3-beta-glucosidase homolog1), a β-1,3-glucanase, showed a contrasting pattern of expression in maize in response to the different microorganisms (Appendix A). This gene was upregulated in the *Zm-B*25*-Fv* and *Zm-Fv* interactions, where *Fv* was present (Log2FC of 2.46 and 2.76, respectively), whereas it was downregulated in the *Zm-B*25 interaction (Log2FC of −2.93). Plant β-1,3-glucanases have a key role in plant–microbe interactions, and several β-1,3-glucanases are included in the pathogenesis-related (PR) group 2 of proteins. This type of β-1,3-glucanase accumulates during a pathogen attack, and some of them have been directly involved in the hydrolysis of pathogen cell walls, since β-1,3-glucans are found in bacteria, metazoa, viruses, and particularly fungi [26]. Thus, the upregulation of *Zm00001eb141540* in maize roots in the presence of the fungus and its downregulation when bacteria are present could suggest that maize perceives the endophytic bacterium *B*25 as a non-threatening microorganism. Further analysis is necessary to demonstrate the role of *Zm00001eb141540* in the *Zm*-*B*25 association. 

Moreover, overrepresented unique GO terms in *Zm-B*25 were related to the establishment of a beneficial plant–bacteria interaction (Figure 3), including carbohydrate metabolic processes (GO:0005975), oligopeptide transport (GO:0006857), and phenylpropanoid metabolic processes (GO:0009698); these compounds have been reported as chemoattractants in plant roots [27]. Plant roots can secrete organic exudates into their surroundings, attracting beneficial microorganisms and establishing a favorable plant–bacteria interaction [28]. Furthermore, the GO term calcium-dependent phospholipase A2 activity (GO:0047498) was exclusively overrepresented in *Zm*-*B*25. A key process in the beneficial plant–bacteria interaction is the increase in calcium flow and reactive oxygen species, leading to lipid oxidation in the cell membrane, indicating a favorable relationship [29]. The GO term lipid oxidation (GO:0034440) appeared between the overrepresented GO terms in *Zm*-*B*25, but also in *Zm*-*Fv.* Quercetin 3-O-glucosyltransferase (GO:0080043) and quercetin 7-O-glucosyltransferase (GO:0080044) activities appeared as overrepresented GO terms only in those interactions where *B*25 was present (*Zm*-*B*25 and *Zm*-*B*25-*Fv*) (Figure 3). Quercetin is a flavonol from the flavonoid subclass that is linked to the establishment of plant interactions with other beneficial organisms, including the arbuscular mycorrhizal association and the symbiosis between legumes and nitrogen-fixing rhizobia [30,31]. Flavonoids also act as communication mediators between PGPRs and plants, suggesting they may play a key role in these interactions [32]. Taking these biological processes together helps explain how maize plants perceive the *B*25 bacterium as a beneficial microorganism. 

### 3.2. Plant–Bacteria Interaction (Zm-B25): B25 Induces the Activation of Pathways Related to Plant Growth, Development, and the Defense Response

Our study revealed the induction of several unique KEGG pathways during the interaction between maize and *B*25 (*Zm-B*25) at seven dpi. These pathways, mainly related to plant growth and development and defense responses, include brassinosteroid biosynthesis; alanine, aspartate, and glutamate metabolism; flavone/flavonol biosynthesis; plant MAPK signaling pathway; and arachidonic acid metabolism. 

Zhang et al. (2022) [33] showed that exogenous application of brassinosteroids (BRs) to maize promotes root growth, enhances stress tolerance, and increases grain yield. Specifically, root development was attributed to the regulation of BRs via auxin, implicating the upregulation of the *ZmWOX5*, *ZmBBM1*, and *ZmBBM2* homologs of *PLT1* and *PLT2,* as well as the auxin transporter genes *ZmPIN2*, *ZmPIN3a*, and *ZmPIN3b*, and the positive auxin response factors (ARF) that promote the initiation of lateral root primordia in *ZmARF7* and *ZmARF19*. None of these genes were induced within the unique DEGs in *Zm-B*25; instead, *Zm00001eb433020* (arftf29-ARF-transcription factor 29) was downregulated (Log2FC = −1.55), and *Zm00001eb026490* (aic1-auxin import carrier1) was upregulated (Log2FC = 1.23). Nonetheless, four genes related to auxin response were identified within the unique EGs in *Zm*-*B*25: *Zm00001eb321090* (saur66—small auxin up RNA66), *Zm00001eb070660* (saur17—small auxin up RNA17), *Zm00001eb321100* (saur67—small auxin up RNA67), and *Zm00001eb218080* (umc2293—auxin responsive protein). These four genes belong to the small auxin-up RNA (*SAUR*) gene family, which itself belongs to three distinct early auxin-responsive gene families [34]. *SAUR* genes are involved in the auxin signaling pathway, and their expression can be induced within 2–5 min by active auxin [34,35]. These observations suggest that the expression of *SAUR* genes, mediated by *B*25, could play a key role in regulating BRs to promote maize growth. 

Amino acids play vital roles in the central metabolism of plants, as they are involved in physiological processes that are directly or indirectly linked to the synthesis of diverse metabolites [36]. Amino acids and their derivatives play prominent roles in plants, including protein synthesis, growth and development, nutrition, and stress responses. Among other amino acids, glutamate, aspartate, and alanine are used in the formulation of biostimulants [37]. Alfosea-Simón et al. (2021) [38] studied the effects of exogenous application of these three amino acids on tomato growth and found a positive effect of aspartic and glutamic acids on plant growth. Foliar application of glutamine (Gln) at the V6, V8, V12, R1 (silk), and R3 (dough) stages in maize hybrid ZD958 impacts grain yield and nutritional quality, enhancing grain yield by 20% under sufficient N, and by 38% under low N [39]. In this work, non-significant differences at an early stage of maize development (7 dpi) were observed between *Zm-B*25 and the control condition (*Zm*) for the evaluated growth parameters (Table 1). This is possibly due to the developmental stage of the evaluated plantlets. Nonetheless, previous fieldwork demonstrated that maize inoculation with the biocontrol agent *B*25 significantly increased plant growth and grain yield at several field sites and crop seasons compared to untreated control (*Zm*) and *Fv*-inoculated (*Zm-Fv*) plants [9]. *B*25 activate glutamate, aspartate, and alanine metabolic pathways, serving as a key mechanism in maize to promote growth and increase grain yield. *B*25-mediated induction of amino acid metabolism at an early growth stage (7 dpi) may remain induced at later maize developmental stages, but further maize transcriptome analysis at later developmental stages will be necessary to confirm this hypothesis. 

Mitogen-activated protein kinases (MAPKs) play essential roles in plant growth and development by regulating both cell differentiation and proliferation [40]. MAPKs are also important signaling modules in eukaryotes, with key roles in regulating responses to four principal environmental stresses (high salinity, drought, extreme temperature, and insect/pathogen infections) via gene expression regulation, plant hormone production, and crosstalk between environmental stresses [41]. In the *Zm-B*25 interaction, 29 DEGs were identified in the overrepresented MAPK plant signaling pathway, most of which related to stress adaptation, defense response, pathogen defense, and late pathogen defense response. All of these DEGs were upregulated, except for *Zm00001eb133330* (Mitogen-activated protein kinase kinase 3).

In addition to the MAPK plant signaling pathway, arachidonic acid metabolism was also identified as an overrepresented KEGG pathway in the *Zm-B*25 interaction, with 5 upregulated DEGs. Arachidonic acid (AA) is an evolutionarily conserved signaling molecule that modulates plant stress signaling networks and acts as an elicitor of pathogen defense response and plant programmed cell death [42,43]. Dye & Bostock (2021) [44] demonstrated that the exogenous addition of AA to the roots of tomato and pepper seedlings protects them against root and crown rot caused by *Phytophtora capsici*. [45] showed that tomato roots treated with AA altered the tomato metabolome with an enrichment of chemical classes and the accumulation of metabolites associated with defense-related secondary metabolism. Among the five upregulated DEGs found in this overrepresented KEGG pathway, *Zm00001eb184300* ((+)-neomenthol dehydrogenase) was the second-most upregulated gene (log2FC = 3.2). The menthone reductase gene *CaMNR1* in *Capsicum annuum* and its ortholog *AtSDR1* in *Arabidopsis thaliana* were observed to regulate plant defenses against a broad spectrum of pathogens positively. Specifically, *CaMNR1*-overexpressing *Arabidopsis* plants exhibited enhanced resistance to the hemibiotrophic pathogen *Pseudomonas syringae* pv. *tomato* DC3000 and the biotrophic pathogen *Hyaloperonospora parasitica* isolate Noco2, whereas a mutation in *AtSDR1* significantly enhanced susceptibility to both pathogens [46]. This suggests that the maize gene *Zm00001eb184300* in the *Zm*-*B*25 interaction could be implicated in preparing plant defense responses against pathogen attack.

The induction of flavone/flavonol biosynthesis detected by KEGG analysis in the *Zm-B*25 interaction could contribute to plant growth and development as well as the defense response, as this plant-specialized secondary metabolite performs many functions in plants, including plant color development, cell growth regulation, attraction to pollinators, and protection against biotic and abiotic stresses [47,48,49]. 

Some other unique GO terms significantly overrepresented in *Zm-B*25 were related to plant defense and growth promotion. These include the cinnamic acid biosynthetic process (GO:0009800), which is involved in the biosynthesis of secondary metabolites related to plant defense and acts as an antioxidant, promoting plant growth [50]; the plant-type secondary cell wall biogenesis (GO:0009834); and plant-type cell wall organization (GO:0009664). The last two GO terms may contribute to optimal cell wall development and the provision of mechanical resistance [51,52]. *Zm00001eb399600* (allene oxide synthase, *aos1*) was a unique upregulated DEG (Log2FC of 2.09) in *Zm-B*25. This defense-related gene is overexpressed in maize plants upon contact with the PGPR *Klebsiella jilinsis* [52].

### 3.3. Plant–Pathogenic Fungus Interaction (Zm-Fv): Fv Turns off DNA Replication and Repair, and Negatively Affects Plant Growth and Development 

The genomic integrity of every organism is constantly challenged by endogenous and exogenous DNA-damaging factors, and maintaining this integrity is crucial to the proper development of all living organisms and the faithful transmission of genetic information from one generation to the next [53,54]. The most important sources of DNA damage are endogenous, including oxidative metabolism, spontaneous hydrolytic reactions, and replication machinery errors. However, environmental factors such as ionizing and UV radiation, and genotoxic biogenic or industrial chemicals, also contribute significantly [55]. Because plants are attached to their substrate, they cannot avoid these stresses. Therefore, proper sensing of DNA damage and the precise activation and functioning of the DNA repair machinery are of foremost importance in preserving genome integrity [56]. In the present work, our results show that the main effect of *Fv* in maize roots is the downregulation of genes involved in DNA replication and maintenance. Both KEGG and GO enrichment analyses (Figure 3 and Figure 4) confirmed the enrichment of unique pathways (motor proteins, ribosome biogenesis in eukaryotes, DNA replication, homologous recombination, base excision repair, and non-homologous end-joining) and GO terms (DNA binding, transmembrane transporter activity, single-stranded DNA helicase activity, DNA replication initiation, and negative regulation of DNA recombination) associated with DNA replication and repair processes. Co-expression network analysis further supports this observation, as it revealed DNA replication and maintenance within the modules with significantly enriched downregulated terms observed only in the *Zm-Fv* condition (Appendix A). Moreover, an independent GO analysis assessing the unique downregulated DEGs (809 out of 1513 DEGs) in this condition confirmed the overrepresentation of the following GO terms related to DNA metabolism: organelle organization, chromosome organization, DNA and nucleic acid metabolic processes, and cellular component organization or biogenesis (Appendix A). 

One key mechanism of DNA repair processes is the DNA Damage Response (DDR). This mechanism activates the DNA repair pathways BER (base excision repair), NER (nucleotide excision repair), and MMR (mismatch repair mechanism) for single-strand breaks, as well as HR (Repair–Homologous Recombination) and NHEJ (Non-Homologous End Joining) for double-strand breaks (DSB) [54,55,56]. We observed that both single-strand and double-strand DNA repair processes were affected by *Fv* in maize roots in the *Zm-Fv* condition. Five DEGs associated with the Non-Homologous End Joining Pathway were downregulated: *Zm00001eb116810* (ATP-dependent DNA helicase 2 subunit KU70), *Zm00001eb371940* (DNA polymerase lambda (POLL)), *Zm00001eb181120* (DNA repair protein RAD50), *Zm00001eb069270* (double-strand break repair protein MRE11), and *Zm00001eb071490* (putative DNA ligase 4). Similarly, a total of 18 DEGs in the Homologous Recombination Pathway were also downregulated, including genes coding for DNA polymerase I A (Zm00001eb431790) and delta (*Zm00001eb201340* and *Zm00001eb002170*), DNA repair proteins (*Zm00001eb255780* and *Zm00001eb181120*), and DNA topoisomerases (*Zm00001eb005010* and *Zm00001eb317420*), as well as 17 DEGs in the DNA Replication Pathway.

*Fv* affected maize root growth and development by turning off the pathways of ribosome biogenesis and motor proteins (Figure 4). From a total of 31 DEGs related to the ribosome biogenesis pathway in the *Zm-Fv* condition*,* only *Zm00001eb332570* (eukaryotic translation initiation factor 6) appeared as upregulated, while the rest were downregulated. Given the central function of ribosomes as the molecular machines responsible for protein synthesis, ribosome biogenesis is tightly linked with plant growth and development [57] and thus also the correct development of plant roots. Moreover, DEGs coding for kinesin (1 DEG), kinesin-like-protein (11), and kinesin-related protein (4) were downregulated, along with two myosin-17 genes. Kinesins and myosins are motor proteins that can move actively along microtubules and actin filaments, respectively, coordinating diverse cellular functions in plants [58]. Kinesins are responsible for unidirectionally transporting various cargos, including membranous organelles, protein complexes, and mRNAs; they also play critical roles in mitosis, morphogenesis, and signal transduction, and contribute directly or indirectly to cell division and cell growth in various tissues (Li et al., 2012) [59]. In contrast, myosin drives cytoplasmic streaming, actin organization, and cell expansion, thus performing supporting roles in plant growth, environmental responses, and defense against pathogens [60]. 

### 3.4. The Tripartite Interaction Zm-B25-Fv: Induction of Defense Response Genes and Plant Growth Development

A common molecular response was found to be derived from the interactions of maize roots with (1) *B*25 (*Zm-B*25); (2) *Fv* (*Zm-Fv*); and (3) both microorganisms (*Zm-B*25-*Fv*). This response included 22,576 core EGs (Figure 1), 1948 core DEGs (Figure 2), 42 core overrepresented GO terms (Figure 3, Appendix A), and 13 core overrepresented KEGG pathways (Figure 4). In contrast, a differential and unique response was observed for each interaction, which was stronger when microorganisms interacted separately with maize than when they interacted simultaneously (i.e., the tripartite interaction). 

The *Zm-B*25-*Fv* tripartite interaction did not present any unique overrepresented KEGG pathways (Figure 4) that could explain the molecular mechanisms involved in the plant response during this interaction, aside from a few unique overrepresented GO terms and unique DEGs. Of the 78 unique DEGs in *Zm-B*25-*Fv*, 34 were upregulated and 44 were downregulated. The top ten upregulated DEGs include four genes that are directly and/or indirectly (orthologous genes) involved in plant resistance against different pathogens. These four genes are *Zm00001eb282040* (glp1—germin-like protein1; log2FC = 4.37); *Zm00001eb339590* (ABC transporter G family member 34; log2FC = 3.87); *Zm00001eb393680* (peroxidase 5; log2FC = 3.18); and *Zm00001eb009380* (LysM-containing Receptor-Like Kinases LYK2; logFC = 2.22). 

When overexpressed in *Arabidopsis*, *Zm00001eb282040* (glp1) improves the resistance to the biotrophic *P. syringae* pv. tomato DC3000 (*Pst*DC3000) and the necrotrophic *Sclerotinia sclerotiorum* pathogens by inducing the expression of JA signaling-related genes [61]. In support of this, the GO term response to jasmonic acid (GO:0009753) was only found as an overrepresented GO term in the *Zm-B*25-*Fv* interaction (Figure 3). *AtABCG34*, the *Arabidopsis* ortholog of *Zm00001eb339590* (ABC transporter G family member 34 [LOC103635032]), mediates camalexin secretion to defend against the necrotrophic pathogens *Alternaria brassicicola* and *Botrytis cinerea* [62]. Compared to the susceptible maize inbred line PH4CV, the resistant inbred maize line 9D207 exhibits significantly induced expression of the phenylpropanoid biosynthesis-related genes *Zm00001eb393680* (peroxidase 5 [LOC103639086]), *Zm00001eb381290* (peroxidase), *Zm00001eb225390* (*EC 3.2.1.21*), *Zm00001eb318150*, *Zm00001eb329520* (*bglB*), *Zm00001eb160830* (*REF1*) and *Zm00001eb158880* (*TOGT1*) when infected by spray-inoculation of an *Exserohilum turcicum* spore suspension [63]. This suggests a possible plant defense role for these phenylpropanoid biosynthesis-related genes. Finally, gene *Zm00001eb009380* ([LOC103633076] Protein LYK2) is a member of the LysM-containing Receptor-Like Kinases (LYKs), which play a key role in the plant defense response. LysM and LYK proteins are the second major class of plant recognition receptors (PRPs) after LRR-RPs/LRR-RLKs, and they play a significant role in the plant immune response [64]. In *A. thaliana,* LYK2 proved to be dispensable for chitin perception and early signaling events, but necessary for enhanced resistance to *B. cinerea* and *P. syringae* induced by flagellin, and for elicitor-induced priming of defense gene expression during fungal infection. LYK2 overexpression in *Arabidopsis* enhances resistance to *B. cinerea* and *P. syringae* and results in increased expression of defense-related genes during fungal infection [65]. Thus, due to the crucial role of LYK2 in *Arabidopsis* for chitin perception, it is possible that Zm00001eb009380 is essential for *Fv* detection, since chitin is a major component of the fungal cell wall [65]. 

Taken together, the upregulation of LYK2, GLP1, peroxidase 5, and ABCG34 suggests that maize plants, when interacting with both a pathogenic fungus and a bacterial control agent, mount a coordinated, multi-layered defense response involving pathogen recognition. First, maize detects the presence of the fungal pathogen by sensing fungal chitin (LYK2), and subsequently, signal amplification is possibly mediated by JA (glp1) and amplified via lignification/ROS-mediated defense responses (Peroxidase 5) that cause the deployment and export of antimicrobial compounds (ABCG34) to allow defense against the pathogen.

Other defense response-related genes were found among the *Zm*-*B*25-*Fv* unique upregulated DEGs, including *Zm00001eb371160* (pme39—pectin methylesterase39), *Zm00001eb415440* (indole-2-monooxygenase-like [LOC103641249]), *Zm00001eb001120* (ATPP2-A13 [LOC100286177]), and *Zm00001eb259950* (NDR1/HIN1-like protein 1 [LOC100284275]). Gene *Zm00001eb371160* (pme39) is a member of the pectin methylesterases (PMEs), which modify pectin, a complex polysaccharide constituent of plant cell walls. PMEs are implicated in various biological processes, including fruit ripening, pathogen defense, and cell wall remodeling [66]. In *A. thaliana*, AtPME17 activity triggers the synthesis of plant defensin 1.2 (PDF1.2) through the jasmonic acid-ethylene signaling pathway, conferring resistance to the fungal pathogen *B. cinerea*, while in wheat, the expression of several PME genes was associated with resistance to *F. graminearum*, the causal agent of head blight [66]. 

*Zm00001eb415440*, an indole-2-monooxygenase-like gene, is involved in the biosynthesis of benzoxazinoids, which are protective and allelopathic compounds found in some plants such as maize, wheat, rye, and other wild and cultivated Poaceae. Benzoxazinoids have been associated with biochemical defense against a variety of biotic stresses, including insect herbivory, microbial pathogens, and competition with other plant species [67]. 

The *A. thaliana* AtPP2-A1 gene ortholog of the maize *Zm00001eb001120* (ATPP2-A13) gene exhibits molecular chaperone and antifungal activities against *F. moniliforme*, *F. solani*, *Rhizoctonia solani*, and *Trichoderma harzianum* [68]. Overexpressing AtPP2-A1 in *A. thaliana* inhibited the phloem-feeding behavior of the aphid *Myzus persicae*, indicating a potential role for PP2 genes in mediating plant defense against insect herbivory [69]. Finally, the gene *Zm00001eb259950* (NDR1/HIN1-like protein 1) is a member of the NHL protein family, involved in plant resistance to biotic and abiotic stresses. In the *Arabidopsis* genome, 45 NHL (NDR1/HIN1-like) genes homologous to NDR1 (non-race-specific disease resistance) or HIN1 (harpin-induced) have been identified, and their roles in pathogen perception have been extensively studied. Overexpression of NHL2 in *Arabidopsis* induced the expression of PR1 (pathogenesis-related gene 1) and light-dependent ‘speck disease-like’ symptoms in the leaves of transgenic plants. Likewise, overexpression of NHL3 increased resistance to *P. syringae* pv. tomato DC3000 [70]. Moreover, the overexpression of the *A. thaliana* genes NHL1 and NHL8 in soybeans enhanced resistance to the pathogen *Heterodera glycines* by activating the jasmonic acid (JA) and ethylene (ET) pathways [71].

Other upregulated unique DEGs found in the *Zm-B*25-*Fv* tripartite interaction relevant to plant defense are: *Zm00001eb153350*, a receptor-like protein 51 (*LRR6*) that can recognize microbial proteins or peptides [72]; *Zm00001eb426640* (nactf65—NAC-transcription factor 65), which is part of the NAC transcription factor family and plays crucial regulatory roles in plant immunity [73]; *Zm00001eb249860*, an actin depolymerizing factor (*adf13*) involved in actin dynamics [74]; *Zm00001eb333050* (α-xylosidase 1 [LOC107548101]), an α-xylosidase that participates in remodeling xyloglucan, involved in the mechanical integrity of the primary cell wall of growing tissues, cell expansion, and seed germination [75] and *Zm00001eb072230*, an S-norcoclaurine synthase 2. Norcoclaurine synthase (NCS) from *Thalictrum flavum* (Tf NCS), *Papaver somniferum* (Ps NCS1 and Ps NCS2), and *Coptis japonica* (Cj PR10A) share substantial identity with pathogen-related 10 (PR10) and Bet v1 proteins [68]. In support of the possible function of these unique upregulated DEGs, the term defense response (GO:0006952) was only found in the *Zm-B*25-*Fv* interaction as an overrepresented GO-term. 

On the other hand, genes *Zm00001eb102960* (Log2FC = −3.37) and *Zm00001eb311640* (Log2FC = −3.64) were found as annotated genes within the top ten most downregulated DEGs. *Zm00001eb311640* is an elongation factor 2 protein that helps ribosomes move along messenger RNA during protein synthesis [76] while *Zm00001eb102960* is an SH3 domain-containing protein 3. Overexpression of SH3P2 in rice variety japonica YY compromised Pib-mediated resistance to *M. oryzae* isolates carrying AvrPib and Pib-AvrPib recognition-induced cell death [77]. Regarding the remaining downregulated DEGs, the genes *Zm00001eb296580* (GTP-binding nuclear protein Ran-2), *Zm00001eb156510* (9DNA ligase 6), *Zm00001eb341090* (parp1—poly(ADP-ribose) polymerase1), *Zm00001eb029490* (TFIID basal transcription factor complex), and *Zm00001eb226110* (DNA repair protein REV1) were related to synthesis and/or DNA repair and transcription. The genes *Zm00001eb101530* (probable galacturonosyl-transferase 4), *Zm00001eb325370* (pdcb44—plasmodesmata callose binding protein44), and *Zm00001eb336480* (rhamnogalacturonan I rhamnosyl-transferase 1) were related to the plant cell wall. The genes *Zm00001eb230490* (mate6—multidrug and toxic compound extrusion6), *Zm00001eb293110* (trps12—trehalose-6-phosphate synthase12), *Zm00001eb251380* (jmj15—JUMONJI-transcription factor 15), *Zm00001eb332340* (mybr2—MYB-related-transcription factor 2), *Zm00001eb182690* (prh37—protein phosphatase homolog37), and *Zm00001eb154830* (type I inositol 1,4,5-trisphosphate 5-phosphatase 1-like) were reported to be associated with salinity, drought, chilling and cold stress. Finally, *Zm00001eb395630* (gpdh3—glucose-6-phosphate dehydrogenase3) was associated with ROS homeostasis; *Zm00001eb203230* (naat1-nicotianamine aminotransferase1) with iron (Fe) acquisition; and *Zm00001eb151610* (inositol-hexakisphosphate kinase/IP6K) with plant defense mechanisms against bacterial, fungal, and viral infections. Overall, despite a considerable number of unique downregulated DEGs related to different mechanisms relevant to maize growth and development, no DEGs stood out in the tripartite interaction. Nevertheless, extended gene network analysis (Appendix A) shows that biological processes involved in plant growth and development, such as gene expression, RNA metabolic process, ribosome biogenesis, translation, actin filament depolymerization, photosynthesis, and glycolysis, may be relevant to the tripartite interaction. This agrees with observations in this work and in the field, where maize seed inoculation with *B*25 results in a growth response and even an increased grain production [9]. Together, the gene interaction network results based on co-expression, using *A. thaliana* orthologous genes and focusing on BPs, support the GO and KEGG findings, as addressed in this Discussion section. 

### 3.5. Synthesis and Conceptual Framework

The transcriptomic analysis of a tripartite interaction model system enabled us to dissect the molecular mechanisms underlying maize root responses to the biocontrol endophytic bacterium *B*25, the fungal phytopathogen *Fv*, and both microorganisms simultaneously (Figure 6). In the plant–bacterium interaction (Figure 6a), maize plants appear to recognize *B*25 as a non-threatening organism; after this recognition, the bacterium induces key mechanisms that promote plant growth, preparing the plant for potential pathogen attack. In the plant–fungus interaction (Figure 6b), the fungal pathogen significantly disrupted the proper function of the plant’s DNA and protein synthesis machinery, thereby inactivating pathways related to these mechanisms. Finally, during the tripartite interaction (Figure 6c), a core of unique genes involved in plant defense responses was found to be essential for successful control of *Fv* infection, along with genes that positively affect plant growth and development.

The present work reveals the molecular mechanisms by which maize interacts with two quite different microbes simultaneously: a beneficial bacterium and a fungal pathogen. These findings provide valuable information that can be useful for the development of future strategies for sustainable maize crop protection. Although we have some previous understanding of how *B*25 triggers direct antagonistic mechanisms that inhibit fungal growth [21], in this work, we started to elucidate general transcriptional responses triggered by the biocontrol agent in the host plant, which help the bacterium to work in conjunction with the plant to mount synergically defense responses against the fungal pathogen. The knowledge gained about these plant responses could lead us to emulate them in the future without the need to add a control agent. 

However, a complete understanding of the molecular mechanisms triggered by microbial inoculation in plants will require further research into the tripartite interaction, as well as the more complex plant–microbe interactions in the rhizosphere. Moreover, since transcript levels may not always reflect protein activity, future proteomic or metabolomic studies will be necessary to confirm the functional effect observed in this work. Metabolomic studies are currently underway to complement, confirm, and/or identify novel pathways related to those described in our transcriptome analysis. 

## 4. Materials and Methods

### 4.1. Microorganisms and Inoculum Preparation

The biocontrol maize endophyte *Bacillus cereus sensu lato* strain *B*25 was obtained from the scientific collection CIIDIR-003 at the CIIDIR-IPN Unidad Sinaloa [13]. The maize pathogenic fungus *Fusarium verticillioides* strain P03, isolated from maize roots exhibiting root rot symptoms from a maize field in El Fuerte Valley, Sinaloa, Mexico [4], were used in this study. Both microorganisms are kept as cryopreserved frozen stocks (−70 °C) at the CIIDIR-IPN Sinaloa Unit. For bacterial inoculum, a powder formulation based on *B*25 spores obtained according to Martínez-Álvarez et al., (2016) [18] was used. The fungal inoculum *Fv* P03 was reactivated from a frozen stock by growing it on PDA plates at 25 °C for 10 days. *Fv* conidia were harvested using sterile distilled water and adjusted to a concentration of 1 × 10^6^ conidia mL^−1^. Harvested conidia were used as fungal inoculum.

### 4.2. Tripartite Assay: Maize–Bacteria–Fungus Interaction

Four conditions were established in this experiment: (1) maize without microorganisms (*Zm*), as a control; (2) maize inoculated with *B*25 (*Zm-B*25); (3) maize inoculated with *Fv* (*Zm-Fv*); and (4) maize inoculated with both *B25* and *Fv* (*Zm-B*25*-Fv*). Commercial maize (*Zea mays* L.) seeds (Garañón hybrid, Asgrow) were surface-disinfected using a hydrothermal treatment [4]. Subsequently, seeds were immersed in a Tween-20 solution (5 drops of Tween-20 per 100 mL of sterile distilled water) and sonicated for 5 min. The Tween solution was then decanted, and seeds were immersed in a 0.75% sodium hypochlorite solution and placed in a thermal bath at 52 °C for 20 min. Finally, the seeds were washed three times with sterile distilled water and allowed to air dry in a laminar flow hood. 

For *Zm-B*25 and *Zm-B*25*-Fv* conditions, disinfected maize seeds were inoculated with *B*25 powder formulation as in Martínez-Álvarez et al. (2016) [18]. Seeds were moistened with 0.4 mL of 1% CMC (*w*/*v*) and then coated with 2 g of powder formulation (1 × 10^9^ spores mL^−1^). The mixture was then stirred until the formulation completely covered the seeds, and the excess formulation was removed. The final inoculum concentration was 1 × 10^6^ CFU of formulated *B*25 spores per seed. For *Zm* and *Zm*-*Fv* conditions, the seeds were coated with the same powder formulation but without *B*25 spores.

The rolled paper towel technique [78] was used, with modifications, to assay the interaction between maize and the microorganisms. Four pieces (1 cm^2^) of sterile filter paper were placed on a 36 × 19.5 cm sterile paper towel previously moistened with sterile distilled water. For *Zm-Fv* and *Zm-B*25-*Fv* conditions, the sterile filter papers were inoculated with 1 × 10^4^ *Fv* conidia; for the *Zm* and *Zm-B*25 conditions, the sterile filter papers were inoculated with sterile distilled water. A single maize seed was placed on each of the four sterile filter papers (four seeds per paper towel), and the paper towels were rolled and placed in sterile plastic bags. Each plastic bag contained three rolls (12 seeds in total) and served as a biological replicate for the fungal biocontrol and plant growth assays. Three plastic bags with three rolls each were placed in an acrylic sterile box and sealed with Micropore tape (Micropore^TM^ surgical tape 1530-1, 3M, CDMX, Mexico) to prevent contamination. Acrylic boxes were then placed in a growth chamber at 28 °C, 60% relative humidity (RH), and a 16:8 h (light:darkness) photoperiod for 7 days. Acrylic boxes were opened once per day in a laminar flow hood to remove excess humidity inside the boxes and to moisten the rolled paper towels with sterile distilled water as needed. Subsequently, the boxes were resealed with micropore tape.

### 4.3. Maize Root Colonization and Plant Protection Mediated by B25 Against Fv

To confirm the protective effect of *B*25 on maize plants against *Fv* and the colonization of maize roots by both microorganisms during the tripartite assay, growth parameters were evaluated in five plants. Superficially disinfected root pieces were then used to re-isolate the microorganisms on LB and PD agar plates. Immediately after harvesting the five plants, fresh weight and shoot and root length were measured using an analytical balance (HR-150AZ, A&D Company, Tokyo, Japan) and a graduated ruler, respectively. Later, roots were superficially disinfected as in Jasim et al. (2014) [79] with modifications. Roots were washed with tap water and cut into 1–2 cm long pieces. Root pieces were submerged in 70% ethanol for 1 min and in 1% sodium hypochlorite for 10 min. Next, the samples were dipped into a 10% (*v*/*v*) Tween-20 solution (Sigma, Cat. P7949) for 1 min and then washed three times with sterile distilled water. Disinfected root pieces were air-dried in a laminar flow hood, longitudinally cut, and placed on LB and PD agar plates, and finally incubated at 30 °C for 24–48 h. 

### 4.4. RNA Extraction and Sequencing

Five out of twelve maize plants from a single bag were randomly selected. Entire root systems were pooled, immediately frozen in liquid nitrogen, and subsequently ground in a sterile mortar and pestle. The root pool from the five plants was used as a biological replicate for RNA extractions. The total RNA of three biological replicates was extracted using TRIzol^®^ reagent (Thermo Fisher Scientific, Cat. No. 15596-026, Waltham, MA, USA). The concentration and quality (260/280 nm ratio) of the total RNA were estimated by spectrophotometry using a Nanodrop 2000c (Thermo Fisher Scientific, Wilmington, DE, USA), and RNA integrity was determined by agarose gel electrophoresis [80]. RNA samples were sent to the Genomic Services Laboratory (LabServGen CINVESTAV-IPN Irapuato, Guanajuato, México) for sequencing. Before processing, the RNA integrity number (RIN) was verified using the Agilent RNA 6000 Nano Kit. All samples had RIN values > 7. Twelve independent libraries, corresponding to three biological replicates from each condition (*Zm*, *Zm-B*25, *Zm-Fv*, and *Zm-B*25*-Fv*), were constructed using the Illumina^®^ TruSeq RNA sample prep v2 kit. Libraries were sequenced in paired-end format (2 × 150 bp) on the Illumina NextSeq 500 platform.

### 4.5. Bioinformatic Analysis

Bioinformatic analyses were performed using the OOREAM computing server at the CIIDIR-IPN Sinaloa Unit. The quality of reads was examined before and after trimming using FastQC v0.11.4 (https://www.bioinformatics.babraham.ac.uk/projects/fastqc (accessed on 30 June 2021)). Raw reads were filtered using Trimmomatic v0.39 [81] with a minimum quality score of 20 and a minimum length of 50 bp. Illumina adapters were also removed during this step. Trimmed reads were mapped to the *Zea mays* B73 v5.0 reference genome [82] using STAR v2.7.0 [83] with default parameters. Gene expression levels in SAM files were estimated from the number of uniquely mapped reads using HTSeq-count v0.11.1 [84]. Raw counts were imported into the R/DESeq2 v1.32 package [85] for differential expression analysis. For principal component analysis (PCA), raw counts from all 12 libraries (4 conditions with three replicates) were normalized using the variance-stabilizing transformation (vst) method. One atypical replicate per condition was identified and removed for further analysis. The remaining eight libraries (4 conditions with two replicates) were then normalized using the median of ratios method to identify differentially expressed genes (DEGs) between the two selected conditions. DEGs were defined as genes with an adjusted *p*-value < 0.01 and |Log2 fold change| > 1. The interaction conditions *Zm-B*25, *Zm*-*Fv*, and *Zm-B*25-*Fv* were individually compared with the control *Zm*, yielding a total of 3 sets of DEGs for each comparison. Next, DEGs from the three interaction conditions (*Zm-B*25, *Zm-Fv*, and *Zm-B*25*-Fv*) were compared in a Venn diagram to identify unique and common DEGs among them.

### 4.6. Gene Function Annotation and Overrepresentation Analysis

Gene ontology (GO) terms were obtained from the Ensembl database [86] using the BioMart tool (https://plants.ensembl.org/index.html (accessed on 31 October 2023)) and assigned to the DEGs of the three interaction conditions. Next, DEGs for each interaction condition were independently subjected to GO and KEGG overrepresentation analyses. 

GO overrepresentation analysis was performed using the R/goseq v1.44 package [87] with the Wallenius method. GO terms with an adjusted *p*-value ≤ 0.05 were considered as significantly overrepresented. 

KEGG overrepresentation analysis was performed in the Database for Annotation, Visualization and Integrated Discovery (DAVID) (https://davidbioinformatics-nih-gov.translate.goog/tools.jsp?_x_tr_sl=en&_x_tr_tl=es&_x_tr_hl=es&_x_tr_pto=tc (accessed on 3 November 2023)). The DEG lists were loaded into the database, and gene identifiers were linked to ENSEMBL_GEN_ID. KEGG pathways with an adjusted *p*-value ≤ 0.05 were considered as significantly overrepresented.

### 4.7. Gene Set Enrichment Analyses: Networks and Clustering

To investigate whether the set of genes from each section and the intersection of maize interactions were associated with specific biological pathways, non-redundant gene identifiers (NRGIs) from *A. thaliana* orthologous genes corresponding to up- and downregulated maize genes were submitted to AgriGO [88], as well as to String [89] and Genemania [90]. The functional enrichment analyses identified GO terms for biological processes (BPs), molecular functions (MFs), and cellular components (CCs). The most significant GO terms and pathways were those with an FDR threshold of 5% and a redundancy score at 0.5. In the network analysis, all gene sets that generated a main network with fewer than 200 nodes were expanded using String. Moreover, the same former gene sets were also submitted to Genemania as a complementary approach to identify enriched pathways. For network clustering based on co-expression, the Markov clustering (MCL) method in StringApp was employed, with an inflation value of 2.5 and STRING interaction scores of 0.4. Only the largest clusters were further used for functional enrichment analysis. Network associations were visualized and analyzed in Cytoscape v3.9.1 [91].

### 4.8. RNA-Seq Validation by qRT-PCR

Quantitative RT-PCR experiments were carried out on the same RNA samples used for RNA-seq. Total RNA was DNase-treated (RQ1 DNase). First-strand cDNA was synthesized from 1 µg of total RNA using SuperScript III reverse transcriptase (Thermo Fisher Scientific, Cat. No.18080-044, Waltham, MA, USA) following the manufacturer’s instructions. qRT-PCR reactions were conducted in a Rotor Gene-Q Real-time PCR system (Qiagen, Cat. No. 9001550, Hilden, Germany); four technical replicates were performed for each of the two biological replicates. Reactions included five μL of SYBR Green master mix (Qiagen, Cat. no. 204074, Hilden, Germany), 1 µM of each primer, 10 ng of cDNA, and RNase-free water for a final volume of 10 μL. For PCR amplification, the thermocycler program included a preheating step at 95 °C for 10 min, followed by 40 cycles of denaturation at 95 °C for 30 s, annealing at 59 °C for 30 s, and extension at 72 °C for 30 s. Dissociation curves were performed at the end of each run to confirm single-product amplifications. Gene-specific primers were designed with Primer3 software (Appendix A); only primers for the housekeeping gene *CDK* [92] were downloaded from the literature. PCR amplification efficiency of the housekeeping and target genes was determined from standard curves generated from serial dilutions of cDNA (1–100 ng). Relative quantification of maize genes was normalized to the housekeeping gene *CDK*, and fold change (FC) values in gene expression were calculated using the comparative threshold cycle method 2^−ΔΔCt^ [93].

### 4.9. Statistical Analysis

Data on plant growth parameters from the tripartite assay samples were assessed for normality using the Kolmogorov–Smirnov test and then analyzed by one-way ANOVA in IBM SPSS Statistics v25. Differences among treatments were determined by Tukey’s test (*p* ≤ 0.05).

## 5. Conclusions

This work advances our understanding of how maize responds at the molecular level when simultaneously exposed to a beneficial bacterium and a fungal pathogen. Beyond the known antagonistic activity of the biocontrol agent, our findings reveal the host’s transcriptional responses that contribute to a coordinated defense against the pathogen. These findings provide a foundation for developing sustainable crop protection strategies that could ultimately mimic these plant responses without the need for microbial inoculants. Future research integrating proteomic and metabolomic approaches will be essential to validate the functional relevance of the identified pathways and deepen our understanding of this tripartite interaction within the rhizosphere.

## Figures and Tables

**Figure 1 plants-14-03661-f001:**
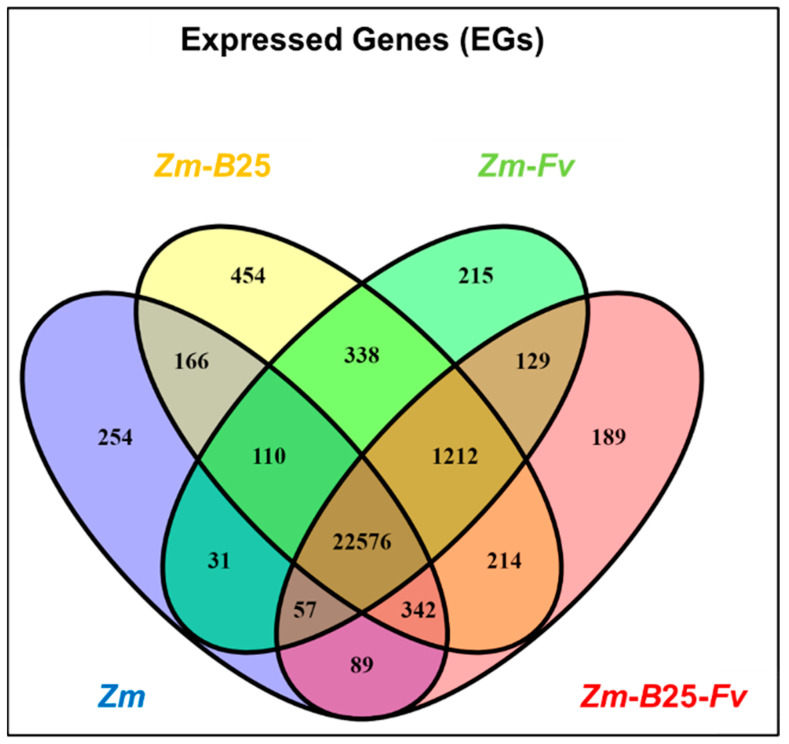
Venn diagram of common and uniquely expressed genes in the transcriptome of maize roots of the tripartite assay. *Zm*: *Zea mays*; *B*25: *Bacillus cereus B*25; *Fv*: *Fusarium verticillioides* P03.

**Figure 2 plants-14-03661-f002:**
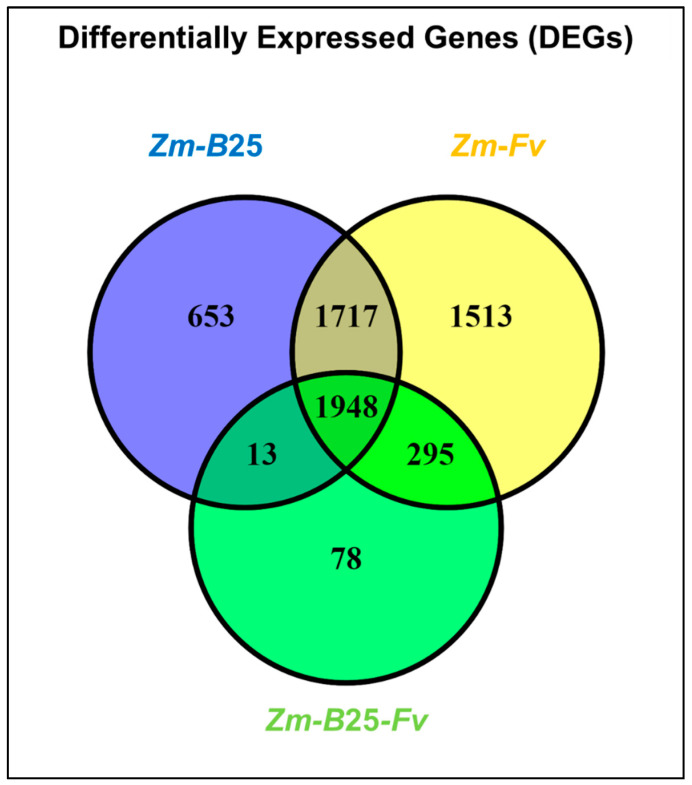
Differentially expressed genes (DEGs) in the tripartite interaction. Venn diagrams of common and unique DEGs between bi- and tri-partite interactions. *Zm*: *Zea mays*; *B*25: *Bacillus cereus B*25; *Fv*: *Fusarium verticillioides* P03.

**Figure 3 plants-14-03661-f003:**
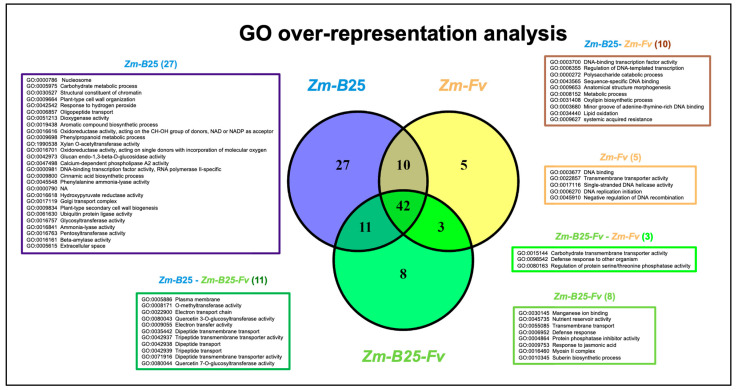
Venn diagram of unique and shared overrepresented Gene Ontology (GO) terms between the interactions. Boxes display the list of the overrepresented GO terms exclusive to one or shared by two interactions. The list of common GO terms (42) shared between the three interactions is shown in Appendix A.

**Figure 4 plants-14-03661-f004:**
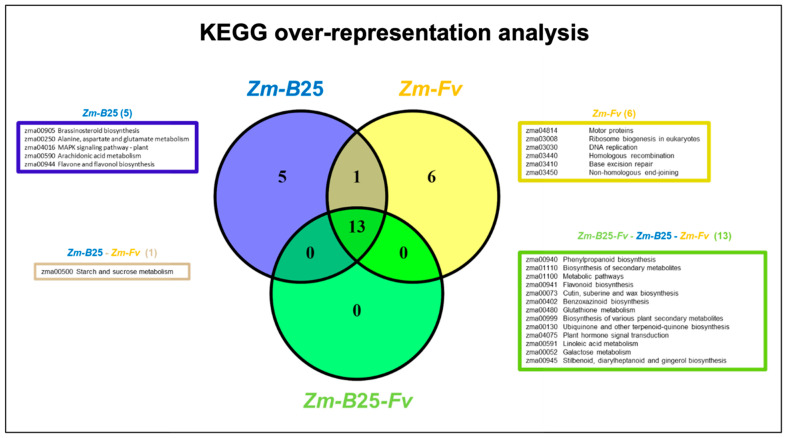
Venn diagram of unique and shared overrepresented KEGG pathways between the interaction conditions. Boxes display the list of overrepresented KEGG terms exclusive to one or shared by two or three interactions.

**Figure 5 plants-14-03661-f005:**
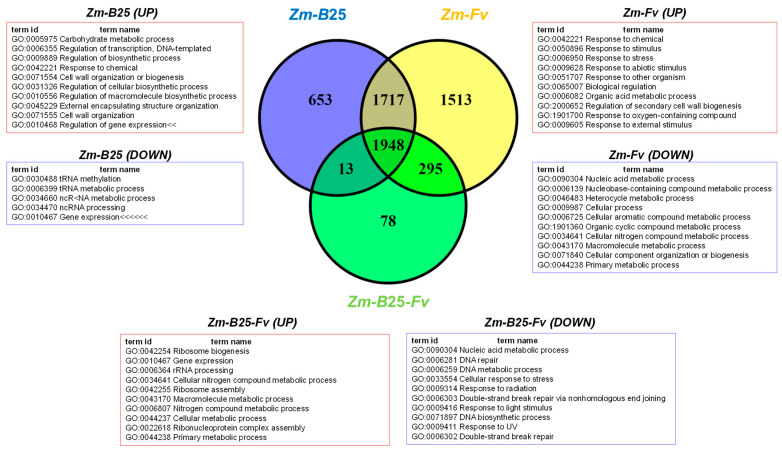
Enriched biological processes emerging after applying the Markov clustering (MCL) based on co-expression.

**Figure 6 plants-14-03661-f006:**
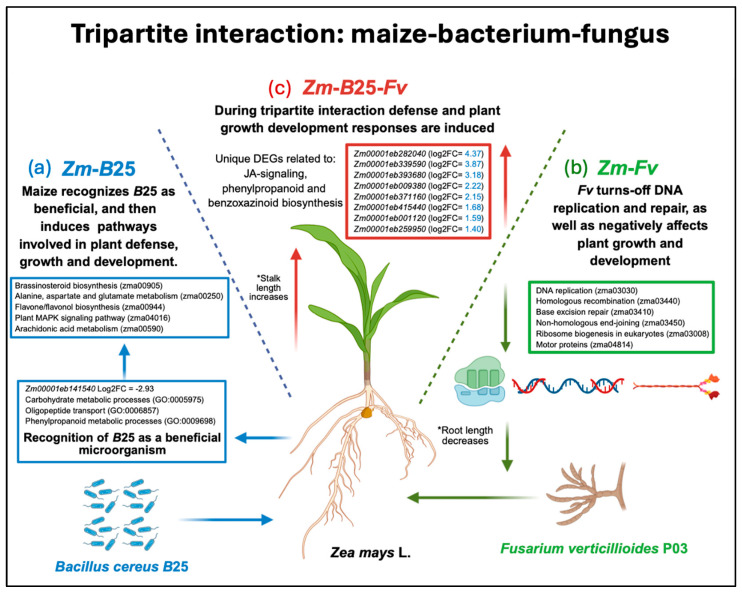
Graphic summary of the main molecular mechanisms involved in maize roots when interacting with the biocontrol endophytic bacterium *B*25, the fungal phytopathogen *Fv*, or both microorganisms (tripartite interaction). (**a**) During the plant–bacterium interaction, *B*25 is recognized by maize as a beneficial microorganism and induces the activation of pathways related to plant growth and development as well as defense responses, preparing the plant for pathogen attack. (**b**) During the plant–fungus interaction, *Fv* turns off key mechanisms of plant DNA replication and repair, as well as pathways of ribosome biogenesis and motor proteins; additionally, it negatively affects plant growth and development, decreasing root length. (**c**) During the maize–bacterium–fungus tripartite interaction, several unique DEGs related to defense response and plant growth development are induced. The most induced maize DEGs in the tripartite interaction are shown in the red box. * Stand for: statistically significant differences between conditions.

**Table 1 plants-14-03661-t001:** Growth parameters of maize plants from the tripartite assay seven days post infection (dpi).

Condition	Length (cm)		Fresh Weight (g)	
	Leaf	Root	Leaf	Root
*Zm*	11.92 ^b^	13.07 ^a^	0.38 ^b^	0.78 ^a^
*Zm-B*25	11.27 ^b^	12.52 ^a^	0.34 ^b^	0.78 ^a^
*Zm*-*Fv*	11.52 ^b^	**9.86 ^b^**	0.34 ^b^	**0.64 ^b^**
*Zm-B*25-*Fv*	**14.28 ^a^**	13.80 ^a^	**0.47 ^a^**	0.77 ^a^

*Zm*: *Zea mays*; *B*25: *Bacillus cereus B*25; *Fv*: *Fusarium verticillioides* P03. Values indicate the average of five plants. Different letters indicate significant differences (Tukey’s test *p* ≤ 0.05). Bold values highlight statistically significant differences as described in the text.

**Table 2 plants-14-03661-t002:** Summary of filtered and mapped reads obtained from the RNA-seq of maize root samples.

Sample	Raw Reads	Filtered Reads (Q > 20)		Unique Mapped Reads	
*Zm*-2	13,742,216	13,534,709	(98.49%)	12,542,341	(91.27%)
*Zm*-3	21,273,970	20,839,981	(97.96%)	17,042,555	(80.11%)
*Zm*-*B*25-1	22,841,141	22,498,524	(98.50%)	19,719,075	(86.33%)
*Zm*-*B*25-2	27,395,643	27,003,885	(98.57%)	24,442,305	(89.22%)
*Zm-Fv*-1	19,155,891	18,880,046	(98.56%)	17,497,335	(91.34%)
*Zm-Fv*-3	18,080,786	17,815,902	(98.54%)	14,513,113	(80.27%)
*Zm*-*B*25-*Fv*-1	18,442,655	18.199,212	(98.68%)	16,495,370	(89.44%)
*Zm*-*B*25-*Fv*-3	21,465,145	21,166,779	(98.61%)	19,373,477	(90.26%)
Average	20,299,681	19,992,380	(98.49%)	17,703,196	(87.28%)

*Zm*: *Zea mays*; *B*25: *Bacillus cereus B*25; *Fv*: *Fusarium verticillioides* P03; Q: Phred quality score. Numbers after the condition name indicate the biological replicate. Filtered reads were mapped to the reference genome of maize B73 v5.

**Table 3 plants-14-03661-t003:** Expressed genes (EGs) in maize roots of the tripartite assay.

Condition	Total EGs	Percentage of the Genome ^a^
*Zm*	23,625	53.33%
*Zm-B*25	25,412	57.36%
*Zm-Fv*	24,668	55.68%
*Zm-B*25*-Fv*	24,808	56.00%
Average	24,628	55.59%

*Zm*: *Zea mays*; *B*25: *Bacillus cereus B*25; *Fv*: *Fusarium verticillioides* P03. ^a^ Refers to the *Zea mays* B73 v5.0 genome (44,303 genes).

**Table 4 plants-14-03661-t004:** Differentially expressed genes (DEGs) in the interaction conditions (vs. control).

Condition	Total DEGs	Up-Regulated	Down-Regulated
*Zm-B*25	4331	3884 (89.68%)	447 (10.32%)
*Zm-Fv*	5473	4076 (74.47%)	1397 (25.53%)
*Zm-B*25*-Fv*	2334	2077 (88.98%)	257 (11.02%)

*Zm*: *Zea mays*; *B*25: *Bacillus cereus B*25; *Fv*: *Fusarium verticillioides* P03.

**Table 5 plants-14-03661-t005:** Comparison of relative expression levels obtained by RNA-seq and qRT-PCR for selected genes from the unique DEGs of the interaction conditions.

Condition	ID ^a^	Gene	Fold Change vs. Control	
			FC RNA-seq	FC qRT-PCR
*Zm*-*B*25-*Fv*	Zm00001eb375460	Glutaredoxin	11.49	4.11
	Zm00001eb229260	Protein kinase domain	2.18	2.24
	Zm00001eb251380	Cyclic nucleotide-binding domain	−2.23	−7.56
	Zm00001eb029490	Gdt1 family	−2.41	−4.37
	Zm00001eb397080	CRAL/TRIO, N-terminal domain	−2.61	−2.74
*Zm*-*B*25	Zm00001eb124940	Multi antimicrobial extrusion protein	2.87	2.92
*Zm*-*Fv*	Zm00001eb041100	Hydroxycinnamoyltransferase 9	13.1	7.3
	Zm00001eb419890	Deoxymugineic acid synthase 6	5.09	5.57
	Zm00001eb285030	Putative cytochrome P450 superfamily protein	7.07	3.72
	Zm00001eb241870	Small auxin up RNA54	9.69	24.98

^a^ Maize reference identifier (*Zea mays* B73 v5.0).

## Data Availability

Raw sequence reads of this study can be downloaded from https://www.ncbi.nlm.nih.gov/bioproject/PRJNA1345406 (accessed on 15 October 2025). The rest of the data related to this work are contained within the article.

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
