# Peer review of "Transcriptional Analysis of a Tripartite Interaction Between Maize (Zea mays, L.) Roots Inoculated with the Pathogenic Fungus Fusarium verticillioides and Its Bacterial Control Agent Bacillus cereus sensu lato Strain B25"

_plants, 2025, doi:10.3390/plants14233661_

Round 1
Reviewer 1 Report
Comments and Suggestions for Authors
Major concerns
- Photos should be added for part 2.1.
- Why the results of the third sample were missing in part 2.2? It should be fully discussed since fewer replicates might affect the results.
- The mapped rates of sequencing reads should be described in part 2.2 as supplementary materials. Have the authors considered to align the unmapped reads to the genomes of the bacteria or fungi? There might also exist DEGs.
- A photo should be added for summary the results of part 2.8.
- Why these genes were validated in part 2.9? The reason should be fully described.
- The authors should reconstruct the discussion part to emphasize that how maize root response to bacteria and fungi, together with why bacteria reduce the harm of fungi. The last part of discussion is quite confusing. The fungi will surely induce defense response genes and 597 plant growth development. What are the differences when bacteria were added?
- The author should be more cautious about gene functions in discussion part since there are no experimental proofs.
Author Response
Reviewer 1
We thank reviewer 1 for the comments and suggestions to improve this manuscript.
- Photos should be added for part 2.1.
R: A supplementary figure has been added to the manuscript and named as Supplementary Figure 2; hence, the previous Supplementary Figure 2 was renamed as Supplementary Figure 3, due to the appearance order.
Due to the extension of the manuscript, we have considered including this new Figure as part of the supplementary material rather than a main figure.
- Why the results of the third sample were missing in part 2.2? It should be fully discussed since fewer replicates might affect the results.
R: For this part, we didn’t miss results; instead, we opted to exclude one biological replication. As a step to confirm biological replicates consistency, we applied a PCA analysis (section 4.5: Lines 831-837). Commonly, this type of analysis is not always employed in transcriptome works; nonetheless, we consider this step as fundamental to obtain robust results, since variation between biological replicates may affect the results and their biological interpretation. Thus, due to variation in one biological replicate per condition, we decided to take it out of the analysis to ensure robust results. Despite the final analysis being performed using two biological replicates for each condition, biological replicates consistency (final replicates), depth of the sequencing, reads quality (>Q20), as well as unique mapping (87.26% on average) remained within the required parameters for transcriptome analysis, thus allowing us to perform the analysis we made on this tripartite interaction in a correct way. Moreover, considering the number of biological replicates and to compensate for this, we used a strict parameter of P-value <0.01 to obtain DEGs.
- The mapped rates of sequencing reads should be described in part 2.2 as supplementary materials. Have the authors considered to align the unmapped reads to the genomes of the bacteria or fungi? There might also exist DEGs.
R: Mapped rates are currently presented in the last column (Unique Mapped Reads) of Table 2 (“Summary of filtered and mapped reads obtained from the RNA-seq of maize root samples”) and briefly described in the text between lines 126-128.
On the other hand, reads that were not mapped to the Zm genome were used to perform alignments against the Fv and B25 genomes, individually. Reads mapped to Fv but not to B25. We must note that the difference with maize is that there is a control treatment (maize alone) against which all other treatments are compared, and DEGs are the result of comparing a treatment with the control. In the case of the Fv reads, they only can be contrasted between the treatments with Fv, specifically Zm-Fv and Zm-Fv-B25. However, the number of reads obtained was not enough to obtain DEGs for Fv.
- A photo should be added for summary the results of part 2.8.
- We appreciate the reviewer’s valuable observation regarding Section 2.8 (“Gene interaction networks based on co-expression”), in which no figure summarizes all the results. We decided not to include a figure summarizing all the co-expression network results in the main text due to the considerable length of the manuscript and the large amount of information generated from the clustering analyses. The Markov clustering (MCL) approach produced multiple functional clusters and associated biological processes across several interaction conditions and intersections. Integrating all these results into a single figure would have required a high degree of simplification, potentially reducing the interpretability and accuracy of the presented information. Instead, we presented these results as a Table (Supplementary Table 7).
- Why these genes were validated in part 2.9? The reason should be fully described.
Previous section 2.9: To validate the RNA-seq data, eleven unique DEGs were selected based on their FC (high and/orlow) from the interactions, and their expression profiles were evaluated by qRT-PCR. As indicated in Table 5, FC values obtained by qRT-PCR analysis displayed similar expression trends as in the RNA-seq data.
To include the reason why validation is suggested in RNA-seq analysis the paragraph was modified to:
To confirm our RNA-seq data analysis, RT-qPCR analysis was conducted to validate the expression patterns of selected genes using a different RNA expression measurement technique. Eleven unique DEGs were selected based on their FC (high and/or low) and the number of reads (³30 reads per biological replicate) from the interactions, and their expression profiles were evaluated by qRT-PCR. As indicated in Table 5, the FC values obtained by qRT-PCR analysis displayed similar expression trends to those in the RNA-seq data, thus confirming the feasibility of the RNA-seq data.
- 1) The authors should reconstruct the discussion part to emphasize that how maize root response to bacteria and fungi, together with why bacteria reduce the harm of fungi.
2) The last part of discussion is quite confusing.
3) The fungi will surely induce defense response genes and 597 plant growth development.
4) What are the differences when bacteria were added?
R: 1) The discussion section of the manuscript was arranged to present the unique molecular responses of maize when interacting with 1) the B25 bacterium, 2) the Fv fungal pathogen, and finally 3) both B25 and Fv. In relation to why bacteria reduce the harm caused by this fungus, our research group has already published several papers (included in the introduction section, L77-85) that report on the mechanisms of antagonism used by B25 against Fv. However, the interaction between maize with both B25 and Fv has not been studied until now.
2) Regarding the discussion being confusing, we previously had a conclusion section that summarizes the main responses on each interaction, including a general scheme, which we decided to include in the last part of the discussion section to help clarify the main events found for each interaction.
3) We do not quite understand this comment. Indeed, Fv induces defense response genes, as demonstrated in this paper and previous reports (see reference 25, Cazares-Alvarez et al., 2024). Nevertheless, Fv does not induce plant growth responses; it is quite the opposite, it has an adverse effect on plant growth and development, as shown in the summary in Figure 6.
4) The effect of protection induced by B25 against Fv in maize plants has also been documented (included in the introduction and discussion sections). Thus, we have previously reported that when bacteria are added to maize plants, Fv cannot cause damage or the damage is decreased. Finally, due to the difficulty in obtaining mRNA from a composite sample that includes two or more organisms, it is not possible to evaluate the three organisms simultaneously. Thus, we focus our analysis on maize, as stated in the introduction section (L86-93).
- The author should be more cautious about gene functions in discussion part since there are no experimental proofs.
R: We agree with the reviewer on this recommendation, considering that we did not validate our gene functions.
To clarify this, we now include phrases like “suggest that”, or we add the verbs “could”, and “might”, due to the limitation for biological interpretation using only results from transcriptomic data. In this regard, another reviewer also suggested adding a perspective to perform proteomic or metabolomic analysis since transcript levels may not always reflect protein synthesis, or metabolic activity. Thus, we added a paragraph addressing this point in the manuscript (L788-791).
Our research group is currently conducting metabolomic analysis of this interaction (maize-B25-Fv) to complement, confirm, and/or find novel pathways regarding those described in our transcriptome analysis.
Reviewer 2 Report
Comments and Suggestions for Authors
- The authors must mention to resource of biocontrol agent (Bacteria), pathogen (Pathogen), and maize seeds in materials and methods in this study.
- "4.2. Tripartite assay: maize-bacteria-fungus interaction":
- The authors must mention to scientific name for maize in this section
- Why are you mentioned to "Tripartite"? Do you use a special seed cultivar of maize?
- Why are you used this (P = 0.05)? Why did not you use (P =>0.05)?
- The authors can not mention to figures in conclusion and must delete it
Author Response
Reviewer 2
We thank Reviewer 2 for the comments and suggestions to improve this manuscript.
- The authors must mention to resource of biocontrol agent (Bacteria), pathogen (Pathogen), and maize seeds in materials and methods in this study.
R: Section 4.1 Microorganisms and inoculum preparation was complemented. This is the new paragraph (L795-799):
“The biocontrol maize endophyte Bacillus cereus sensu lato strain B25 was obtained from the scientific collection CIIDIR-003 at the CIIDIR-IPN Unidad Sinaloa [13] and the maize pathogenic fungus Fusarium verticillioides strain P03, isolated from maize roots exhibiting root rot symptoms from a maize field in El Fuerte Valley, Sinaloa, Mexico [4] were used in this study”.
The source of maize seeds is already described in section 4.2:
Tripartite assay: maize-bacteria-fungus interaction (L809-811): Commercial maize seeds (Garañón hybrid, Asgrow) were surface-disinfected using a hydrothermal treatment.
- "4.2. Tripartite assay: maize-bacteria-fungus interaction":
- The authors must mention to scientific name for maize in this section
R: The scientific name of maize was added in this section:
“Commercial maize (Zea mays L.) seeds (Garañón hybrid, Asgrow) were surface-disinfected using a hydrothermal treatment.”
- Why are you mentioned to "Tripartite"? Do you use a special seed cultivar of maize?
R: Tripartite refers to the interaction of three organisms, in this work these are: maize, the bacterium and the fungal pathogen.
- Why are you used this (P = 0.05)? Why did not you use (P =>0.05)?
R: We found four typos and corrected the p value to: p ≤ 0.05 throughout the text.
- The authors can not mention to figures in conclusion and must delete it
R: We agree with this comment. The conclusion section was deleted from the manuscript (since it is optional), and instead, to present as a concluding remark the integrative scheme and overall biological interpretation we introduced a discussion section entitled Synthesis and Conceptual Framework. This suggestion was also made by another reviewer highlighting keeping Figure 6 as part of the discussion.
Reviewer 3 Report
Comments and Suggestions for Authors
This article presents an interesting and well-designed study on the molecular basis of interactions between maize (Zea mays), a fungal pathogen (Fusarium verticillioides), and a bacterial biocontrol agent (Bacillus cereus sensu lato B25). The topic is relevant and important, addressing complex plant–microbe relationships that are key for sustainable agriculture and the development of biological plant protection methods. The use of transcriptomic analysis (RNA-seq) to study a tripartite interaction is a strong point of the paper, as most previous research has focused only on dual interactions (plant–pathogen or plant–beneficial microbe).
Transcriptomic studies are very informative for understanding plant–pathogen–endophyte interactions, but metabolomic analyses are also needed for a complete picture. Changes at the transcript level do not always correspond directly to biological activity because of translational and post-translational regulation.
I find the manuscript very good, but there are several aspects that could be improved, for example:
-
Add a schematic figure summarizing how Bacillus cereus B25 influences maize defense and growth pathways during co-inoculation with Fusarium verticillioides.
-
Expand the discussion of highly upregulated genes such as ABC transporter G family member 34, LYK2, and peroxidase 5, explaining their possible roles in plant defense.
-
Mention that transcript levels may not always reflect protein activity, and suggest future proteomic or metabolomic studies to confirm these functional effects.
-
End the Discussion with a short paragraph about how the results could be applied to developing microbial biocontrol products and improving sustainable crop protection strategies.
Recommendation: Minor revision.
Author Response
Reviewer 3
This article presents an interesting and well-designed study on the molecular basis of interactions between maize (Zea mays), a fungal pathogen (Fusarium verticillioides), and a bacterial biocontrol agent (Bacillus cereus sensu lato B25). The topic is relevant and important, addressing complex plant–microbe relationships that are key for sustainable agriculture and the development of biological plant protection methods. The use of transcriptomic analysis (RNA-seq) to study a tripartite interaction is a strong point of the paper, as most previous research has focused only on dual interactions (plant–pathogen or plant–beneficial microbe).
Transcriptomic studies are very informative for understanding plant–pathogen–endophyte interactions, but metabolomic analyses are also needed for a complete picture. Changes at the transcript level do not always correspond directly to biological activity because of translational and post-translational regulation.
R: We thank the comments and suggestions made by the reviewer for improving this work.
I find the manuscript very good, but there are several aspects that could be improved, for example:
- Add a schematic figure summarizing howBacillus cereus B25 influences maize defense and growth pathways during co-inoculation with Fusarium verticillioides.
R: The manuscript included previously a scheme that summarizes the tripartite interaction regarding each interaction (1: maize-bacteria, 2: maize-fungus and 3: maize-bacteria-fungus) at the conclusion section (Figure 6). The journal´s article structure allows authors to add a conclusion section as optional, given that this section appeared after the material and methods section interrupting the connection between the discussion and the conclusion, we decided to eliminate the conclusion section, and instead, we added this at the end of the Discussion section including Figure 6 (Last section: Synthesis and conceptual framework).
The way we addressed the discussion and constructed this figure was by finding in each one of the three interactions unique DEGs, GO terms and Pathways. We could not find unique pathways for maize during co-inoculation (maize-bacteria-fungus). Nonetheless, we observed several unique highly upregulated genes and directed our discussion and scheme towards this set of unique genes.
- Expand the discussion of highly upregulated genes such asABC transporter G family member 34, LYK2, and peroxidase 5, explaining their possible roles in plant defense.
R: We thank the reviewer for this suggestion, and we added a paragraph in the discussion section integrating the possible roles of these genes (L656-663).
“Taken together, the upregulation of LYK2, GLP1, peroxidase 5, and ABCG34 suggests that maize plants, when interacting with both a pathogenic fungus and a bacterial control agent, mount a coordinated, multi-layered defense response involving pathogen recognition. First, maize detects the presence of the fungal pathogen by sensing fungal chitin (LYK2), and subsequently, signal amplification is possibly mediated by JA (glp1) and amplified via lignification/ROS-mediated defense responses (Peroxidase 5) that cause the deployment and export of antimicrobial compounds (ABCG34) to allow defense against the pathogen.”
- Mention that transcript levels may not always reflect protein activity and suggest future proteomic or metabolomic studies to confirm these functional effects.
R: A paragraph has been added (L788-791):
“Moreover, since transcript levels may not always reflect protein activity, future proteomic or metabolomic studies will be necessary to confirm the functional effect observed in this work. Metabolomic studies are currently under way to complement, confirm, and/or find novel pathways regarding those described in our transcriptome analysis.”
We expect that the research on metabolomic analysis currently being conducted in our research group in this interaction (maize-B25-Fv) will provide confirmation on data obtained by KEGG analysis in the maize-B25 and maize-Fv interactions of certain metabolic pathways and could lead us to better interpret the results on the tripartite interaction.
- End the Discussion with a short paragraph about how the results could be applied to developing microbial biocontrol products and improving sustainable crop protection strategies.
R: A paragraph has been added (L779-784):
“These findings provide valuable information that can be useful for the development of future strategies for sustainable maize crop protection. Although we have some previous understanding of how B25 triggers direct antagonistic mechanisms that inhibit fungal growth [21], in this work, we started to elucidate general transcriptional responses triggered by the biocontrol agent in the host plant, which help the bacterium to work in conjunction with the plant to mount synergically defense responses against the fungal pathogen. The knowledge gained about these plant responses could lead us to emulate them in the future without the need for the addition of a control agent.”
By taking all reviewer’s suggestions, we decided to delete the conclusion section to avoid missing the flow of ideas between the end of the discussion section, with the Materials and Method section followed by the conclusion, and instead we included the integrative scheme and final considerations in the new Synthesis and Conceptual Framework discussion section at the end of the Discussion.
Round 2
Reviewer 1 Report
Comments and Suggestions for Authors
The authors have addressed most of the comments. The manuscript could be accepted after grammar checking.
Author Response
The authors have addressed most of the comments. The manuscript could be accepted after grammar checking.
R: We had previously used an English editing service (Improvence), although the manuscript suffer changes during final writing and revision. Thus, we have conducted grammar, punctuation and construction check of the whole manuscript text as requested by the reviewer. Changes are highlighted in yellow throughout the text.
Reviewer 2 Report
Comments and Suggestions for Authors The authors must add a part for conclusionAuthor Response
The authors must add a part for conclusion.
R: we have added a conclusion section after the Material and Methods Section as indicated in the instructions for authors. The conclusion is highlighted in yellow. Also, we have conducted grammar check of the whole manuscript and yellow highlighting was used to show all changes in grammar and punctuation.
Round 3
Reviewer 2 Report
Comments and Suggestions for Authors
NA
Author Response
These changes were already made in the second round of reviews.